# The Roles and Regulatory Mechanisms of Tight Junction Protein Cingulin and Transcription Factor Forkhead Box Protein O1 in Human Lung Adenocarcinoma A549 Cells and Normal Lung Epithelial Cells

**DOI:** 10.3390/ijms25031411

**Published:** 2024-01-24

**Authors:** Daichi Ishii, Yuma Shindo, Wataru Arai, Takumi Konno, Takayuki Kohno, Kazuya Honda, Masahiro Miyajima, Atsushi Watanabe, Takashi Kojima

**Affiliations:** 1Department of Thoracic Surgery, Sapporo Medical University School of Medicine, Sapporo 060-8556, Japan; d10660348@yahoo.co.jp (D.I.); shindo.y@sapmed.ac.jp (Y.S.); w.arai@sapmed.ac.jp (W.A.); samuraiguts2@gmail.com (K.H.); mmiyajima@sapmed.ac.jp (M.M.); atsushiw@sapmed.ac.jp (A.W.); 2Department of Cell Science, Institute of Cancer Research, Sapporo Medical University School of Medicine, Sapporo 060-8556, Japan; tkonno@sapmed.ac.jp (T.K.); kohno@sapmed.ac.jp (T.K.)

**Keywords:** cingulin (CGN), forkhead box protein O1 (FOXO1), HDAC inhibitor, angulin-1/LSR, claudin-2, claudin-4, TGF-β, cellular metabolism, lung adenocarcinoma, malignancy, human lung epithelial cells

## Abstract

Tight junction (TJ) protein cingulin (CGN) and transcription factor forkhead box protein O1 (FOXO1) contribute to the development of various cancers. Histone deacetylase (HDAC) inhibitors have a potential therapeutic role for some cancers. HDAC inhibitors affect the expression of both CGN and FOXO1. However, the roles and regulatory mechanisms of CGN and FOXO1 are unknown in non-small cell lung cancer (NSCLC) and normal human lung epithelial (HLE) cells. In the present study, to investigate the effects of CGN and FOXO1 on the malignancy of NSCLC, we used A549 cells as human lung adenocarcinoma and primary human lung epithelial (HLE) cells as normal lung tissues and performed the knockdown of CGN and FOXO1 by siRNAs. Furthermore, to investigate the detailed mechanisms in the antitumor effects of HDAC inhibitors for NSCLC via CGN and FOXO1, A549 cells and HLE cells were treated with the HDAC inhibitors trichostatin A (TSA) and Quisinostat (JNJ-2648158). In A549 cells, the knockdown of CGN increased bicellular TJ protein claudin-2 (CLDN-2) via mitogen-activated protein kinase/adenosine monophosphate-activated protein kinase (MAPK/AMPK) pathways and induced cell migration, while the knockdown of FOXO1 increased claudin-4 (CLDN-4), decreased CGN, and induced cell proliferation. The knockdown of CGN and FOXO1 induced cell metabolism in A549 cells. TSA and Quisinostat increased CGN and tricellular TJ protein angulin-1/lipolysis-stimulated lipoprotein receptor (LSR) in A549. In normal HLE cells, the knockdown of CGN and FOXO1 increased CLDN-4, while HDAC inhibitors increased CGN and CLDN-4. In conclusion, the knockdown of CGN via FOXO1 contributes to the malignancy of NSCLC. Both HDAC inhibitors, TSA and Quisinostat, may have potential for use in therapy for lung adenocarcinoma via changes in the expression of CGN and FOXO1.

## 1. Introduction

Cancer is a leading cause of death worldwide. In particular, lung cancer is the most common cause of cancer-related death around the world and about 1.6 million people die of it every year [1]. A group of histological subtypes, non-small cell lung cancers (NSCLC) accounting for about 85% of lung cancers, are the most common subtypes [2]. Although approximately 85% of lung cancers are related to tobacco smoking, for adenocarcinoma, the number of nonsmokers is rising and the number of cases in never-smokers is also rising, especially among women and in East Asia [3].

Tight junction (TJ) components affect several signaling and transcriptional pathways that regulate gene expression, cell proliferation, cell migration, cell differentiation, and apoptosis [4,5]. Claudins (CLDNs), which are known bicellular TJ proteins, are frequently dysregulated in various cancers, so they are promising biomarkers for diagnosis or targets for treatment [6]. Claudin-2 (CLDN-2) and claudin-4 (CLDN-4) are leaky-type tight junction proteins, and their overexpression increases the tumorigenesis of some types of cancer cells [7,8,9,10,11]. CLDN-4 affects the malignancy of various cancers, including lung cancers [10,11]. Tricellular TJ protein angulin-1/Lipolysis-stimulated lipoprotein receptor (LSR) is an important molecule of tricellular contacts in the epithelial barrier of normal cells and contributes to the malignancy of various cancer cells [12]. The loss of angulin-1/LSR promotes the malignancy via EGF-dependent CLDN-2 and TGF-β-dependent cell metabolism in human lung adenocarcinoma [13]. Cingulin (CGN) acts as a scaffold protein to connect TJ strands and actin in TJ [14], but has recently been implicated in oncogenesis. The loss of CGN affects the malignancy of various cancers [15,16,17].

On the other hand, the forkhead transcription factors of the forkhead box protein (FOXO) subfamily are known to be shared components among pathways that regulate diverse cellular functions such as differentiation, metabolism, proliferation, and survival [18,19]. Furthermore, the FOXO subfamily consists of four members (FOXO1, FOXO3, FOXO4, and FOXO6) in mammals, and these four FOXO transcription factors bind to target genes as monomers or heterodimers and control cell fate under different conditions [20]. Numerous histopathological studies have demonstrated an association between a low expression of FOXO and increased cancer metastasis [21,22,23]. Forkhead box protein (FOXO1) is downregulated in human NSCLC tissues and the silencing of FOXO1 promotes the proliferation, migration, and invasion of NSCLC cells in vitro, whereas the overexpression of FOXO1 inhibits the migration and invasion [24]. The loss of FOXO1 leads to alterations in organization of the tight junction protein occludin in the intestine [25]. However, the relationship between FOXO1 and CGN is unknown.

Many target gene alterations have been identified in lung cancer, including EGFR, KRAS, BRAF, PI3K, MEK, and HER2. EGFR plays an important role in regulating normal cell growth, apoptosis, and other cell functions [26]. EGFR mutations occur in 10–20% of patients not of east Asian descent with NSCLC and in about 40% of Asian patients, mostly in adenocarcinoma, younger women and girls, and never-smokers [26,27,28]. The blockade of EGFR with specific tyrosine kinase inhibitors (TKIs) can produce dramatic tumor responses in NSCLC [29,30,31]. However, tumor recurrence is common and it is important for the improvement of NSCLC outcomes to study new drugs.

Histone deacetylases (HDACs) are epigenetic regulators that regulate the histone tail, chromatin conformation, protein–DNA interaction, and even transcription [32,33]. To date, 18 mammalian HDACs have been identified. These are characterized into four classes: class I HDACs (HDACs 1, 2, 3, and 8), class II (HDACs 4, 5, 6, 7, 9, and 10), class IV (HDAC 11), and class III (sirtuin family: sirt1-sirt7) [30,31]. Furthermore, HDACs are increased in malignant cells and are closely associated with the acquisition of a malignant phenotype in carcinogenesis [32]. Overexpression of HDAC1 and 3 hase been reported to be correlated with a poor prognosis in NSCLC [34,35].

HDAC inhibitors are epigenetic regulators and can be used as a new major treatment for various cancers. To date, four HDAC inhibitors as anti-cancer drugs (vorinostat, romidepsin, belinostat, and panobinostat) have been approved by the United States Food and Drug Administration (US FDA) [32]. In addition, more HDAC inhibitors are being developed. Trichostatin A (TSA) is a potent and specific inhibitor of class I and class II HDACs and exerts antitumor effects in a variety of cancers, including breast, bladder, and head and neck cancer [36,37,38]. Quisinostat (JNJ-26481585) is a novel second-generation HDAC inhibitor with the highest potency against HDAC1 and an orally bioavailable anticancer drug [39]. In NSCLC, we reported that TSA and Quisinostat prevented cell proliferation and migration via the downregulation of CLDN-2 and upregulation of angulin-1/LSR, and TSA induced cellular metabolism with or without TGF-β [40]. However, the effects of the HDAC inhibitors of CGN, CLDN-4, and FOXO1 are unknown in NSCLC and in normal HLE cells.

In this study, we investigated the role and regulation of TJ protein CGN and transcription factor FOXO1 in the malignancy of human lung adenocarcinoma compared to normal lung epithelial cells. It was found that the downregulation of CGN induced malignancy via upregulation of MEK-dependent CLDN-2, cell metabolism and cell migration in human lung adenocarcinoma A549 cells. Furthermore, the downregulation of FOXO1 induced malignancy via the downregulation of CGN and upregulation of CLDN-2, CLDN-4, and cell proliferation and cell metabolism in A549 cells. The HDAC inhibitors TSA and Quisinostat increased CGN and angulin-1/LSR in A549.

## 2. Results

### 2.1. Expression and Localization of CGN in Lung Adenocarcinoma

To investigate changes in the distribution and expression of CGN during the carcinogenesis of human lung adenocarcinoma, immunohistochemical staining for CGN was performed using paraffin-embedded sections of lung cancer tissues (six different adenocarcinomas: four papillary, two invasive). In normal lung tissues, CGN was detected in peripheral bronchial epithelium, but it was not detected in alveolar epithelium (Figure 1A,E). In atypical adenomatous hyperplasia (AAH)-like lesions, CGN was strongly detected at the membranes (Figure 1B,E). In papillary and invasive adenocarcinomas, CGN was highly expressed at the membranes and it was detected in the cytoplasm of some cancer cells (Figure 1B–E). In some invasive adenocarcinomas, CGN was faintly expressed at the membranes of some cancer cells (Figure 1D,E).

### 2.2. HDAC Inhibitors TSA and Quisinostat Increased CGN and Angulin-1/LSR in Lung Adenocarcinoma Cell Line A549

HDAC inhibitors have a potential therapeutic role for NSCLC. We previously reported that the class I and II HDAC inhibitors TSA and Quisinostat (JNJ) have potential for use in therapy for lung adenocarcinoma via changes in the expression of angulin-1/LSR and CLDN-2 [39]. To investigate the effects of the HDAC inhibitors TSA and Quisinostat (JNJ) on CGN, human lung adenocarcinoma cell line A549 cells were treated with them after pretreatment with various signaling inhibitors. In Western blot analysis, TSA and Quisinostat (JNJ) increased the expression of CGN and angulin-1/LSR (Figure 2A). Pretreatment with U0126 (MEK1/2 inhibitor) prevented the upregulation of CGN and angulin-1/LSR induced by TSA and Quisinostat (JNJ), whereas pretreatment with WZ4003 or GF109203X did not affect the upregulation (Figure 2A).

### 2.3. Knockdown of CGN Increased Expression of CLDN-2 in A549 Cells

It is known that CLDN-2 also contributes to the cell proliferation and chemoresistance of lung adenocarcinoma [41,42]. The downregulation of angulin-1/LSR induces the malignancy of lung adenocarcinoma via EGF-dependent CLDN-2 and TGF-β-dependent cell metabolism [13]. To investigate how CGN contributed to the malignancy of lung adenocarcinoma, A549 cells were transfected with the siRNAs of CGN after pretreatment with HDAC inhibitors and various signaling inhibitors. The knockdown of CGN increased the expression of angulin-1/LSR, CLDN-2, acetylated tubulin, phosphorylated mitogen-activated protein kinase (pMAPK), and phosphorylated adenosine monophosphate-activated protein kinase (pAMPK) in Western blot analysis (Figure 2B). Pretreatment with U0126 (MEK1/2 inhibitor) prevented the upregulation of CGN and angulin-1/LSR induced by the knockdown of CGN under treatment with TSA and Quisinostat (JNJ) (Figure 2B). Immunocytochemical staining revealed that CGN was not found in the control, whereas CLDN-2 was detected at the membranes and in the cytoplasm (Figure 2C). Treatment with TSA and Quisinostat (JNJ) increased the expression of CGN at the membranes (Figure 2C). The knockdown of CGN increased the expression of CLDN-2 at the membranes and in the cytoplasm with or without HDAC inhibitors (Figure 2C).

### 2.4. Knockdown of FOXO1 Increased Expression of CLDN-4 and Decreased Expression of CGN in A549 Cells

The suppression of FOXO1 functions promotes the malignancy, metastasis, and EMT in NSCLC [43]. To investigate the relationships of CGN and transcription factor FOXO1 in lung adenocarcinoma, A549 cells were transfected with siRNA-CGN or siRNA-FOXO1 with or without Quisinostat (JNJ). In Western blot analysis, the knockdown of FOXO1 decreased the expression of CGN under treatment with Quisinostat (JNJ) and increased the expression of CLDN-4 under treatment with or without Quisinostat (JNJ), while treatment with Quisinostat (JNJ) increased the expression of CGN and FOXO1 (Figure 3A). Immunocytochemical staining revealed that the knockdown of FOXO1 decreased the expression of CGN at the membranes under treatment with Quisinostat (JNJ), as well as the knockdown of CGN (Figure 3B). TGF-β is involved in the promotion of malignancy, which induces EMT, in lung adenocarcinoma [42]. The changes of CLDN-4 expression induced by the knockdown of CGN and FOXO1 were compared to those caused by treatment with TGF-β by using WT cells and single cell cloning cells (D9 and F8). Western blot analysis showed that the knockdown of FOXO1, but not CGN, increased CLDN-4 expression, which was also upregulated by treatment with TGF-β (Figure 3C). Treatment with TGF-β increased cell migration, but not cell proliferation (Appendix A). In this experiment, the knockdown of CLDN-4 by the siRNA did not affect cell migration or cell proliferation (Appendix A).

### 2.5. Knockdown of CGN Induced Cell Migration but Not Cell Proliferation and Knockdown of FOXO1 Induced Cell Proliferation but Not Cell Migration in A549 Cells

To investigate whether the knockdown of CGN and FOXO1 affected cell proliferation and cell migration in lung adenocarcinoma, A549 cells were transfected with CGN or FOXO1 and cell cycle assay indicated cell proliferation. A cell migration assay showed that the knockdown of CGN, but not FOXO1, induced cell migration compared to the control (Figure 4A,B). In the cell cycle assay, the G0/G1 phase was remarkably decreased and the S and G2/M phases were increased by the knockdown of FOXO1, but not CGN, compared to the control (Figure 4C).

### 2.6. Knockdown of CGN and FOXO1 Induced Cell Metabolism in A549 Cells

In our investigation into the impact of CGN and FOXO1 knockdown on cell metabolism in lung adenocarcinoma, we utilized CGN and FOXO1 siRNA for the knockdown and subsequently analyzed the cell metabolism parameters. The knockdown of CGN increased the cell metabolism, evident from changes in baseline oxygen consumption rates (OCR), maximal OCR, spare respiratory capacity (SRC), and ATP production, while the knockdown of FOXO1 increased the baseline OCR (Figure 5A–D).

### 2.7. HDAC Inhibitors Upregulated Expression of CGN and CLDN-4 and Knockdown of CGN and FOXO1 Upregulated Expression of CLDN-4 in Normal Human Lung Epithelial Cells (HLE Cells)

In normal HLE cells, HDAC inhibitors were found to upregulate the expression of angulin 1/LSR, TRIC, CLDN 2, and 7, while downregulating the expression of CLDN 1 [40]. The knockdown of LSR in normal HLE cells does not affect the expression of other tight junction molecules [13].

To investigate the effect of HDAC inhibitors on CGN expression in HLE cells, the cells were treated with TSA and Quisinostat (JNJ). Western blot analysis revealed that treatment with TSA and Quisinostat (JNJ) led to an increase in the expression of CGN, LSR, CLDN-4, and acetylated tubulin in normal HLE cells (Figure 6A).

Furthermore, to investigate the effects of CGN and FOXO1 knockdown on CLDN-4 expression in normal HLE cells, the cells were transfected with CGN, FOXO1, and CLDN-4. Western blot analysis demonstrated that the knockdown of CGN and FOXO1 resulted in an increased expression of CLDN-4 compared to the control (Figure 6B). Transfection with siRNA-CLDN-4 downregulated the CLDN-4 expression induced by the knockdown of CGN and FOXO1 (Figure 6B).

## 3. Discussion

In this present study, our findings indicate that the downregulation of CGN and FOXO1 leads to malignant cell proliferation, cell migration, and altered cell metabolism by upregulating CLDN-2 or CLDN-4 through the MAPK/AMPK pathways in human lung adenocarcinoma A549 cells (Figure 7). Furthermore, the HDAC inhibitors TSA and Quisinostat were observed to induce CGN and angulin-1/LSR via the MAPK pathway in A549 cells (Figure 7).

It is known that tight junction (TJ) proteins play a crucial role not only in maintaining barrier function, but also in tumorigenesis through the regulation of gene expression and modulation of signaling pathways in various cancers [44,45]. CGN acts as a scaffold protein, connecting TJ strands and actin in TJ [14]. The loss of CGN has been implicated in the malignancy of various cancers, including salivary duct adenocarcinoma [15], osteosarcoma [16], and colorectal cancer [17]. In the present study, immunohistochemistry revealed a strong detection of CGN at the membranes in AAH-like lesions and in papillary and invasive adenocarcinomas. Interestingly, in some invasive adenocarcinomas, CGN expression was faint at the membranes of certain cancer cells.

CLDNs are major tight junction proteins that mediate cellular polarity and differentiation. The leaky-type TJ protein CLDN-2 has been identified as a potential target for cancer therapy in endometrioid endometrial adenocarcinoma [9]. In lung adenocarcinoma, CLDN-2 exhibits high expression through an EGFR/MEK/ERK/c-Fos pathway, and its overexpression has been linked to increased proliferation [7,46]. In previous studies, it has been demonstrated that the knockdown of angulin-1/LSR in Calu-3 cells induces barrier disruption through the upregulation of CLDN-2 and influences cellular metabolism via AMPK activation in the airway epithelium [47]. The downregulation of angulin-1/LSR induces malignancy via the upregulation of EGF-dependent CLDN-2 and cell metabolism via TGF-β signaling that induces EMT in lung adenocarcinoma [13].

On the other hand, the upregulation of CLDN-4 also plays a role in the malignancy of various cancers, including gastrointestinal cancers [11], breast cancers [48], and lung cancers [10,11]. A recent study demonstrated the involvement of CLDN-4 in modulating cell proliferation and chemotherapeutic sensitivity in gastric cancer [11]. In lung cancer, a comparison between adenocarcinoma and small-cell lung cancer (SCLC) revealed a significant difference only in CLDN-2 expression. However, when comparing adenocarcinoma with squamous cell carcinoma, significant differences were observed in the expression of CLDN-3, CLDN-4, and CLDN-7 [10]. These findings suggest that TJ proteins such as CGN, CLDN-2, and CLDN-4 may indeed be potential targets for therapy in lung adenocarcinoma.

FOXO1 is a forkhead transcription factor of the FOXO subfamily and a shared component among pathways regulating diverse cellular functions such as differentiation, metabolism, proliferation, and survival [18]. Furthermore, the upregulation of FOXO1 affects the malignancy of various cancers, including non-small-cell lung cancer (NSCLC) [49,50]. In NSCLC, overexpressing FOXO1 could inhibit the proliferation and metastasis of A549 cells [49]. In the present study, the knockdown of FOXO1 decreased CGN, increased CLDN-4, and induced cell proliferation and cell metabolism. The downregulation of FOXO1 may contribute the malignancy of NSCLC via the downregulation of CGN.

The reprogramming of cellular metabolism is recognized as one of the hallmarks of cancer [51]. The Warburg effect, a type of metabolic reprogramming, refers to the metabolic switch in cancer cells from oxidative phosphorylation to aerobic glycolysis, initiated by the impairment of mitochondrial respiration [52]. Tumor cells depend on mitochondrial metabolism, and aerobic glycolysis and alterations of intracellular and extracellular metabolism affect gene expression, cellular differentiation, and the tumor microenvironment [53]. Previous studies have demonstrated that the knockdown of angulin-1/LSR induces aberrant cellular metabolism in Calu-3 and A549 cells [47,49]. Additionally, it has been reported that TSA, but not Quisinostat, may suppress malignancy through the modulation of cellular metabolism in lung adenocarcinoma [40]. Consequently, this study aimed to explore how the downregulation of CGN or FOXO1, leading to the upregulation of CLDN-2 or CLDN-4 in A549 cells, affects cellular metabolism, as measured by the baseline oxygen consumption rate (OCR). The downregulation of both CGN and FOXO1 induced alterations in cellular metabolism, affecting OCR, maximal respiration, spare respiratory capacity (SRC), coupling efficiency, proton leak, and ATP production. These findings suggest that CGN and FOXO1 may contribute to the malignancy of lung adenocarcinoma through the modulation of cellular metabolism.

In A549 cells, we previously reported that the HDAC inhibitors TSA and Quisinostat prevented cell proliferation and cell migration via the downregulation of CLDN-2 and upregulation of angulin-1/LSR, and that TSA induced cellular metabolism with or without TGF-β [40]. In the present study, TSA and Quisinostat increased CGN, as well as angulin-1/LSR, via the MAPK pathway in A549. In normal HLE cells, the HDAC inhibitors increased CGN and CLDN-4.

In a protein–protein interaction map, CGN can be seen to directly interact with CLDN-2, CLDN-4, and angulin-1/LSR in human tissues (Appendix A). In the present study, CGN may have interacted with CLDN-2, CLDN-4, and angulin-1/LSR in A549 cells and normal HLE cells.

In The Cancer Genome Atlas (TCGA), a low expression of FOXO1 decreased the survival rate not only in lung adenocarcinoma, but also lung cancer (all), whereas a high expression of CGN decreased the survival rate in lung cancer (all) (Appendix A). On the other hand, a high expression of CLDN-4 decreased the survival rate in lung adenocarcinoma (Appendix A).

In conclusion, in human lung adenocarcinoma, the downregulation of CGN induces malignancy via the upregulation of MEK-dependent CLDN-2 and cell metabolism and cell migration, and the downregulation of FOXO1 induces malignancy via the downregulation of CGN and CLDN-4 and cell proliferation. The downregulation of both CGN and FOXO1 induces cellular metabolism. The HDAC inhibitors TSA and Quisinostat prevent cell proliferation via the upregulation of FOXO1 and migration via the upregulation of CGN. Thus, TSA and Quisinostat may have potential for use in therapy for lung adenocarcinoma via changes in the expression of CGN and FOXO1.

## 4. Materials and Methods

### 4.1. Ethics Statement

The research protocol on human subjects was reviewed and approved by the ethics committee of the Sapporo Medical University School of Medicine. Written informed consent was obtained from each patient who participated in the study. All experiments were performed in accordance with approved guidelines and the Declaration of Helsinki.

### 4.2. Antibodies and Reagents

Trichostatin A (TSA) was obtained from Sigma-Aldrich (St. Louis, MO, USA) and Quisinostat (JNJ-26481585) was obtained from Selleck Chemicals (Houston, TX, USA). A TGF-β receptor type 1 inhibitor (EW-7197) and EGF receptor inhibitor (AG-1478) were obtained from Cayman Chemical (Ann Arbor, MI, USA). A JNK inhibitor (SP600125), mitogen-activated protein kinase kinase (MAPKK) (U0126), and PKCα inhibitor (GF109203X) were from the Calbiochem-Novabiochem Corporation (San Diego, CA, USA). A selective inhibitor of NUAK1 and NUAK2 (WZ4003) was from Selleck Chemicals (Houston, TX, USA). Rabbit polyclonal anti-cingulin (CGN) was from Bethyl Laboratories, Inc. (Montgomery, TX, USA) and mouse monoclonal CGN (G6) was from Santa Cruz Biotechnology (Dallas, TX, USA). Rabbit polyclonal anti-tricellulin (TRIC); anti-claudin (CLDN)-1, -2, -4 antibodies; and mouse monoclonal CLDN-2 were from Zymed Laboratories (San Francisco, CA, USA). Rabbit polyclonal anti-lipolysis-stimulated lipoprotein receptor (LSR) antibodies were obtained from Novus Biologicals (Littleton, CO, USA). Rabbit polyclonal anti-FOXO1, anti-phosphorylated MAPK (pMAPK), and AMPK (pAMPK) antibodies were from Cell Signaling Technology (Danvers, MA, USA). Mouse monoclonal anti-acetylated tubulin and a rabbit polyclonal anti-actin antibody were obtained from Sigma-Aldrich (St. Louis, MO, USA). Alexa 488 (green)-conjugated anti-rabbit IgG and Alexa 594 (red)-conjugated anti-mouse IgG antibodies and Alexa 594 (red)-conjugated phalloidin were obtained from Molecular Probes, Inc. (Eugene, OR, USA). HRP-conjugated polyclonal goat anti-rabbit IgG was obtained from Dako A/S (Glostrup, Denmark). The ECL Western blotting system was obtained from GE Healthcare UK, Ltd. (Buckinghamshire, UK). FITC-dextran (FD-4, MW 4.0 kDa) was obtained from Sigma-Aldrich Co. (St. Louis, MO, USA).

### 4.3. Immunohistochemical Analysis

Human lung tissues were procured from 8 patients diagnosed with lung adenocarcinoma (2 invasive, 2 papillary, 2 acinar, and 2 solid) who underwent lobectomy at Sapporo Medical University. All patients provided informed consent, and the study received approval from the ethics committees of the institution. We used 8 tissues around adenocarcinoma tissues as normal lung tissues. The tissues were embedded in paraffin after fixation with 10% formalin in PBS. Briefly, 5-μm-thick sections were dewaxed in xylene, rehydrated in ethanol, and heated with Vision BioSystems Bond Max using ER2 solution (Leica, Wetzlar Germany) in an autoclave for antigen retrieval. Endogenous peroxidase was blocked by incubation with 3% hydrogen peroxide in methanol for 10 min. After two washes with Tris-buffered saline (TBS), sections were preblocked with Block Ace for 1 h. Following TBS wash, sections were incubated with anti-CGN (1:400) antibodies for 1 h, washed thrice in TBS, and incubated with a Vision BioSystems Bond Polymer Refine Detection Kit DS9800. After three TBS washes, a diamino-benzidine tetrahydrochloride working solution was applied. Finally, the sections were counterstained with hematoxylin. A negative control was performed by replacing the first antibodies with normal rabbit serum.

### 4.4. Cell Line Culture and Treatment

A549 cells, derived from human lung adenocarcinoma, were obtained from both RIKEN Bio-Resource Center (Tsukuba, Japan) and the American Type Culture Collection (ATCC, Rockville, MD, USA). The cells were cultured in Dulbecco’s modified Eagle’s medium (DMEM, Nacalai Tesque, Inc.; Kyoto, Japan), supplemented with 10% dialyzed fetal bovine serum (FBS; Invitrogen, Carlsbad, CA, USA), 100 U/mL penicillin, 100 μg/mL streptomycin, and 50 μg/mL amphotericin-B. For cultivation, the cells were plated on 35 and 60 mm culture dishes coated with rat tail collagen (500 μg of dried tendon/mL in 0.1% acetic acid) and incubated in a humidified 5% CO_2_ incubator at 37 °C. Some cells underwent treatment with various agents, including 100 ng/mL TGF-β1, 10 μM U0126, 10 μM SP600125, 10 μM WZ4003, 10 μM GF109203X, 10 μM AG-1478, 1 or 10 μM TSA, or 1 or 10 μM JNJ, for a duration of 24 h.

### 4.5. Isolation and Culture of Human Lung Epithelial (HLE) Cells

Human lung tissues were acquired from patients undergoing lobectomy at Sapporo Medical University Hospital, diagnosed with adenocarcinoma. Informed consent was obtained from all patients, and the study received approval from the ethics committee of Sapporo Medical University.

The human lung tissues were minced into pieces 2 to 3 mm3 in volume and washed with PBS containing 100 U/mL penicillin and 100 μg/mL streptomycin (Lonza Walkersville, Walkersville, MD, USA) three times. These minced tissues underwent digestion in 10 mL of Hanks’ balanced salt solution with 0.5 μg/mL DNase I and 0.04 mg/mL Liberase (Roche, Basel, Switzerland) with incubation in O_2_ gas mixed with 5.2% CO_2_ at 37 °C for 20–30 min. The dissociated tissues were filtered through a 300 μm mesh followed by a 40 μm mesh (Cell Strainer, BD Biosciences, San Jose, CA, USA). Stromal cells were removed by filtration, and the remaining cells were backwashed and collected as epithelial cells. After centrifugation at 1000× *g* for 2 min, isolated cells were cultured in bronchial epithelial basal medium (BEBM, Lonza Walkersville) containing 4% fetal bovine serum (FBS) (CCB, Nichirei Bioscience, Tokyo, Japan) and supplemented with BEGM^®^ SingleQuots^®^ (Lonza Walkersville, including 0.4% bovine pituitary extract, 0.1% insulin, 0.1% hydrocortisone, 0.1% gentamicin, amphotericin-B [GA-1000], 0.1% retinoic acid, 0.1% transferrin, 0.1% triiodothyronine, 0.1% epinephrine, 0.1% human epidermal growth factor), 100 U/mL penicillin, 100 μg/mL streptomycin, and 50 μg/mL amphotericin-B on 35 and 60 mm culture dishes (Corning Glass Works, Corning, NY, USA) or in 35 mm glass wells (Iwaki, Chiba, Japan), coated with rat tail collagen (500 μg of dried tendon/mL of 0.1% acetic acid). The cells were incubated with or without 10% FBS in a humidified 5% CO_2_:95% air incubator at 37 °C. Some cells were subjected to treatment with 100 ng/mL TGF-β1, 100 ng/mL EGF, 10 μM U0126, 10 μM SP600125, 10 μM WZ4003, 10 μM PKCα inhibitor, 1 or 10 μM TSA, or 1 or 10 μM JNJ.

### 4.6. RNA Interference and Transfection RNA Interference and Transfection

siRNA duplex oligonucleotides against CGN and CLDN-4 were synthesized by Thermo Fisher Scientific (Waltham, MA, USA). An siRNA duplex oligonucleotide against FOXO1 (#6242) was obtained from Cell Signaling Technology (Danvers, MA, USA). The sequences were as follows: siRNA of CGN (sense: 5′-CCCACCAUGCUGCAGUUCAAAUCAA-3′; antisense: 5′-UUGAU-UUGAACUGCAGCAUGGUGGG-3′), siRNA of CLDN-4 (sense: 5′-GCAACAUUGUCACCUCGCAGACCAU-3′; antisense: 5′-AUGGUCUGCGAGGUGACAAUGUUGC-3′). At 24 h after plating, the cells were transfected with 100 nM siRNAs of CGN, FOXO1, and CLDN-4 using Lipofectamine^TM^ RNAiMAX Reagent (Invitrogen) for 48 h. A scrambled siRNA sequence (BLOCK-iT Alexa Fluor Fluorescent, Invitrogen) was employed as control siRNA.

### 4.7. Immunocytochemical Staining

A549 cells and HLE cells cultured in 35 mm glass-coated wells (Iwaki, Chiba, Japan) were fixed with a mixture of cold acetone and ethanol (1:1) at –20 °C for 10 min. Following a rinse in PBS, the cells were incubated overnight at 4 °C with anti-CGN and anti-CLDN-2 antibodies (1:100). Subsequently, the cells underwent three washes in PBS. Alexa Fluor 488 (green)-conjugated anti-rabbit IgG and Alexa Fluor 592 (red)-conjugated anti-mouse IgG (Invitrogen) were used as secondary antibodies. The specimens were observed and captured using an Olympus IX 71 inverted microscope (Olympus Co.; Tokyo, Japan) and a confocal laser scanning microscope (LSM510; Carl Zeiss, Jena, Germany).

### 4.8. Western Blot Analysis

The cultured cells were scraped from 60 mm dishes containing 400 μL of buffer (1 mM NaHCO_3_ and 2 mM phenylmethylsulfonyl fluoride), collected in microcentrifuge tubes, and sonicated for 10 s. The protein concentrations in the samples were determined using a BCA protein assay regent kit (Pierce Chemical Co.; Rockford, IL, USA). Aliquots of 15 μL of protein/lane for each sample were separated by electrophoresis in 5–20% SDS polyacrylamide gels (Wako, Osaka, Japan) and transferred electrophoretically to a nitrocellulose membrane (Immobilon; Millipore Co.; Bedford, UK). The membrane was saturated with blocking buffer (25 mM Tris, pH 8.0, 125 mM NaCl, 0.1% Tween 20, and 4% skim milk) for 30 min at room temperature and then incubated overnight at room temperature with the following primary antibodies: anti-LSR (1:1000), anti-TRIC (1:1000), anti-CLDN-1, -2, -4 (1:1000), anti-CGN (1:1000), anti-FOXO1 (1:1000), anti-pMAPK (1:1000), anti-pAMPK (1:1000), anti-Ac-tubulin (1:1000), and anti-actin antibodies (1:1000). Subsequently, the membrane was incubated with HRP-conjugated anti-mouse and anti-rabbit IgG antibodies at room temperature for 1 h. Immunoreactive bands were detected using an ECL Western blotting system.

### 4.9. Migration Assay

After plating A549 cells onto 35 mm dishes, they were cultured until reaching confluence. At 48 h after the transfection with siRNAs, a wound was created in the cell layer using a plastic pipette tip (P100), and then at 8 h after wound healing, the length of the wound was subsequently measured using a microscope imaging system (Olympus, Tokyo, Japan).

### 4.10. Cell Cycle Assay

A549 cells cultured in 35 mm dishes were collected with 0.05% Trypsin-EDTA and washed once with PBS. Subsequently, the cells were suspended in 1 mL of ice-cold 70% ethanol and incubated for a minimum of 3 h at −20 °C. Afterward, the cells were washed once with PBS, followed by another wash with 200 μL of Muse Cell Cycle reagent (Merck Millipore, Burlington, MA, USA). The cells were then incubated for 30 min at room temperature in the dark. The cell cycle analysis was conducted using a Muse*^®^* Cell Analyzer in accordance with the manufacturer’s instructions.

### 4.11. XF96 Extracellular Flux Measurements

Mitochondrial respiration was evaluated using an XF96 Extracellular Flux Analyzer (Aligent, Santa Clara, CA, USA). A549 cells were plated on XF96 plates at a density of 20,000 cells/well after 24 h of incubation in either DMEM medium with high glucose or glucose-free medium. One day before the experiment, sensor cartridges were hydrated with XF calibrate solution (pH 7.4) and incubated at 37 °C in a non-CO_2_ incubator for 24 h. Baseline measurements of mitochondrial respiration (OCR) were recorded prior to the sequential injection of the following inhibitors: 1 μM oligomycin, an ATP synthase inhibitor; 2 μM FCCP, a mitochondrial respiration uncoupler; and 1 μM antimycin A and rotenone, mitochondrial electron transport blockers. Oligomycin was applied first to estimate the proportion of basal OCR coupled to ATP synthesis. Following oligomycin application, FCCP was used to further determine the maximal glycolysis pathway capacity.

### 4.12. Data Analysis

The presented results in each set are representative of a minimum of three independent experiments. Data are expressed as means ± SEM. The group differences were assessed using a one-way analysis of variance (ANOVA), followed by a post hoc test and an unpaired two-tailed Student’s *t*-test.

## Figures and Tables

**Figure 1 ijms-25-01411-f001:**
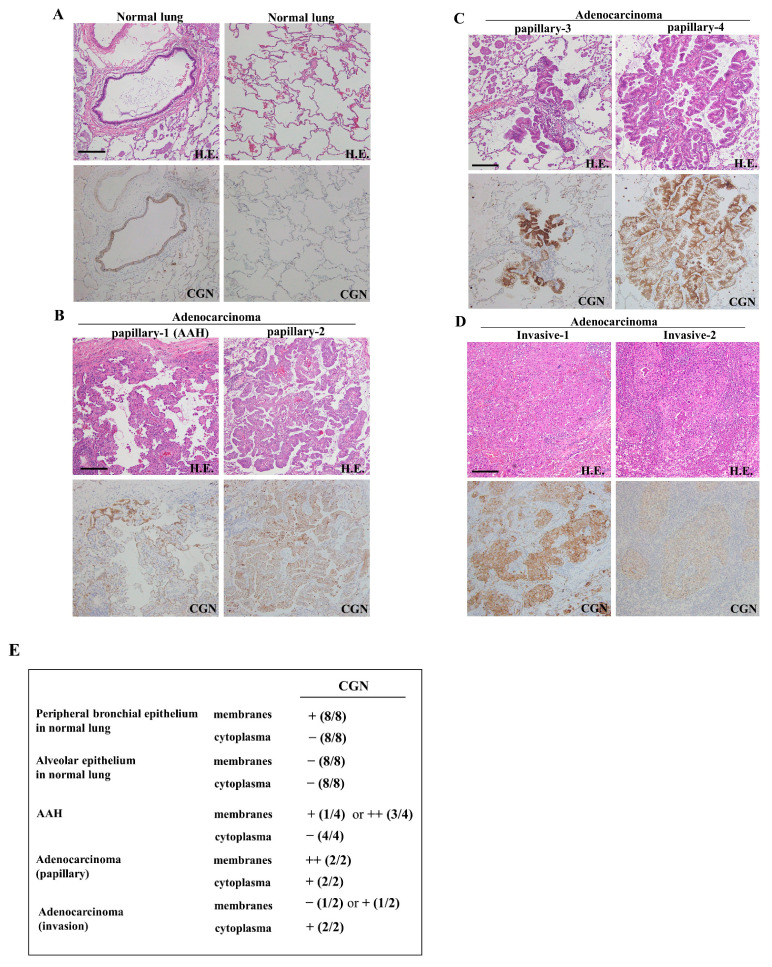
Expression and distribution of CGN in human lung adenocarcinoma. Hematoxylin and eosin (H.E.) and immunohistochemical staining for CGN (brown) in normal (peripheral bronchial, alveolar) and lung adenocarcinoma (papillary, invasive) tissues. Scale bars: 100 μm. (**A**) Normal lung tissues, (**B**,**C**) papillary adenocarcinomas, (**D**) invasive adenocarcinomas. (**E**) Table of expression and localization of CGN. The extent of the positively stained area is indicated by the percentage of atypical cells showing positive IHC staining in the observed area: − (0%), + (1–25%), and ++ (25–50%).

**Figure 2 ijms-25-01411-f002:**
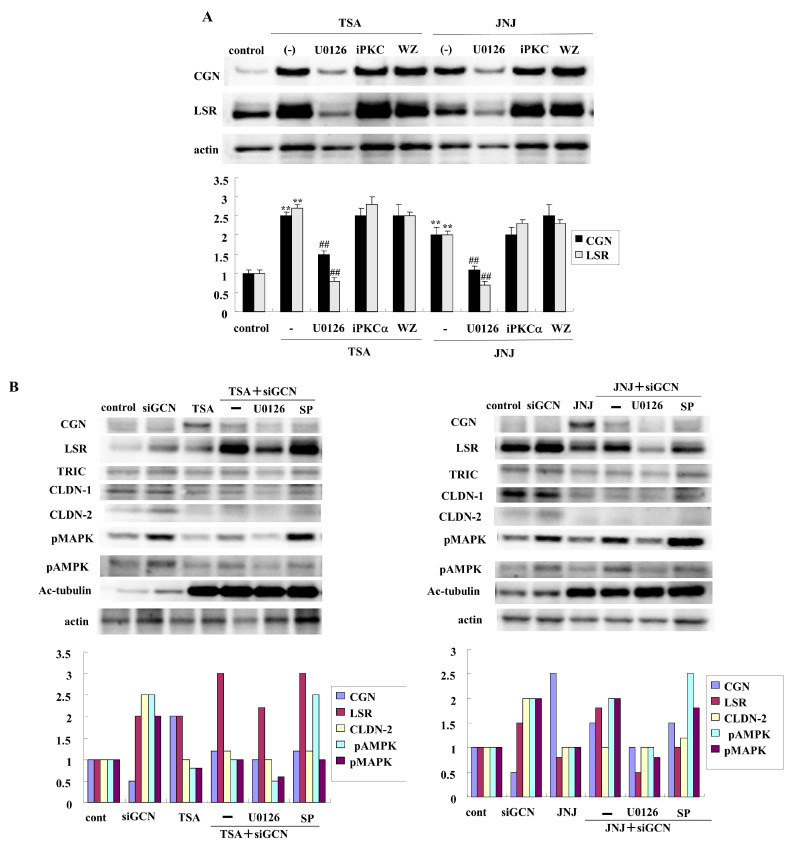
(**A**) Western blotting for angulin-1/LSR, CGN, and actin in A549 cells treated with the HDAC inhibitors TSA and Quisinostat at 10 μM. The corresponding levels are shown as bar graphs. ** *p* < 0.01 vs. control, ## *p* < 0.01 vs. TSA or JNJ treatment. (**B**) Western blotting for angulin-1/LSR, CGN, TRIC, CLDN-1, CLDN-2, pMAPK, pAMPK, Ac-tubulin, and actin in A549 cells treated with the HDAC inhibitors TSA and Quisinostat at 10 μM with and without the knockdown of CGN. The corresponding levels are shown as bar graphs. (**C**) Images of immunocytochemical staining of CGN (green) and CLDN-2 (red) in A549 cells treated with a HDAC inhibitor at 10 μM with and without the knockdown of CGN. Scale bars: 20 μm.

**Figure 3 ijms-25-01411-f003:**
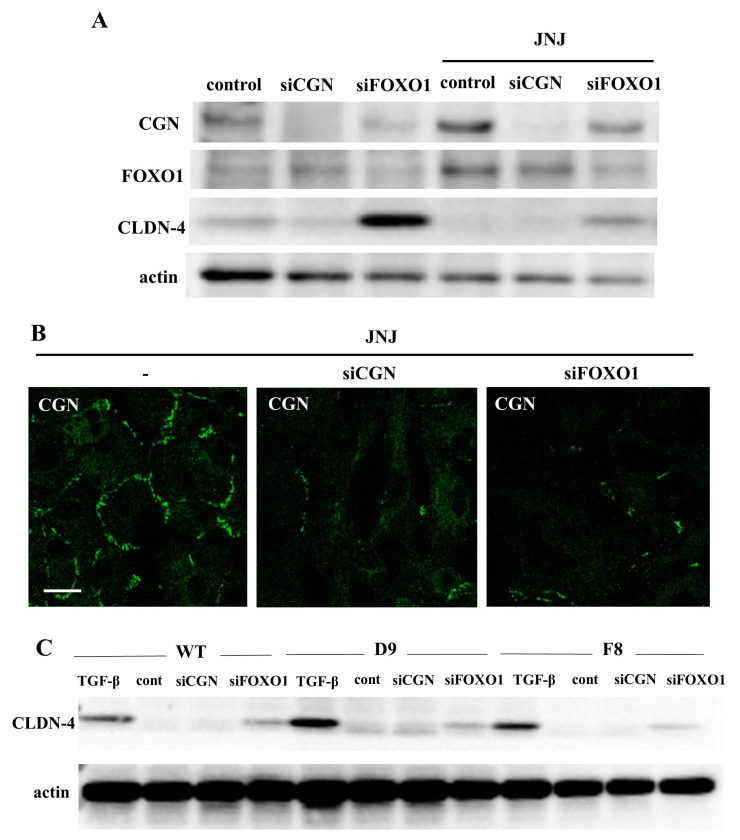
(**A**) Western blotting for CGN, FOXO1, CLDN-4, and actin in A549 cells treated with the HDAC inhibitor Quisinostat at 10 μM with and without the knockdown of CGN and FOXO1. (**B**) Images of immunocytochemical staining of CGN in A549 cells treated with Quisinostat at 10 μM and the knockdown of CGN or FOXO1. Scale bars: 20 μm. (**C**) Western blotting for CLDN-4 and actin in A549 cells treated with TGF-β with and without the knockdown of CGN and FOXO1.

**Figure 4 ijms-25-01411-f004:**
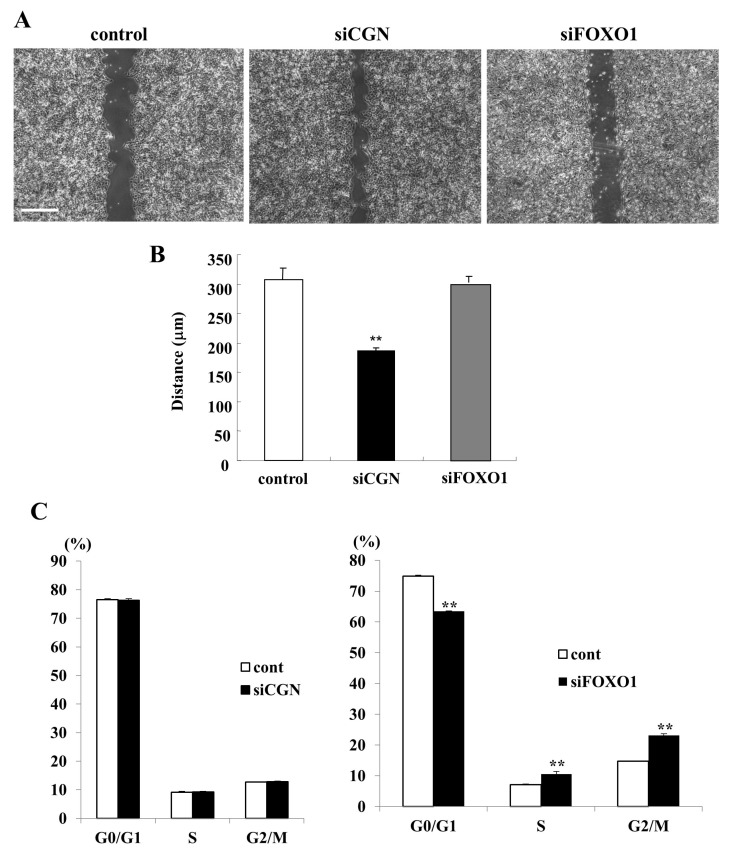
(**A**,**B**) Images of scratch wound assay in A549 cells with the knockdown of CGN or FOXO1. The distance is shown as a bar graph. Scale bars: 200 μm. ** *p* < 0.01 vs. control. (**C**) Cell cycle assay of A549 cells with the knockdown of CGN or FOXO1. The results are shown as bar graphs. ** *p* < 0.01 vs. control.

**Figure 5 ijms-25-01411-f005:**
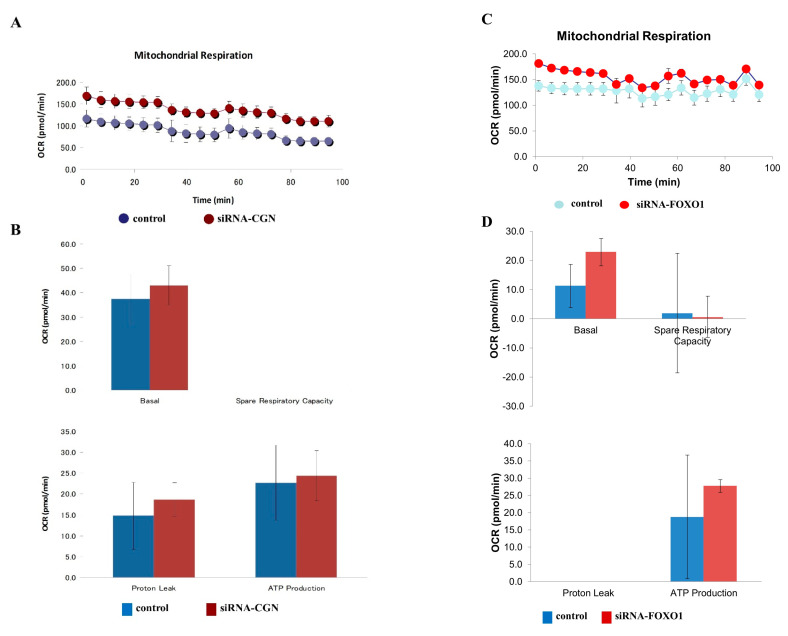
Mitochondrial stress tests using Seahorse Bioscience XF Analyzers for A549 cells with the knockdown of CGN (**A**,**B**) or FOXO1 (**C**,**D**). The baseline oxygen consumption rate (OCR), maximal respiration, non-mitochondrial oxygen consumption, coupling efficiency, proton leak, spare respiratory capacity (SRC), ATP production, and the percentage of SRC are shown as bar graphs.

**Figure 6 ijms-25-01411-f006:**
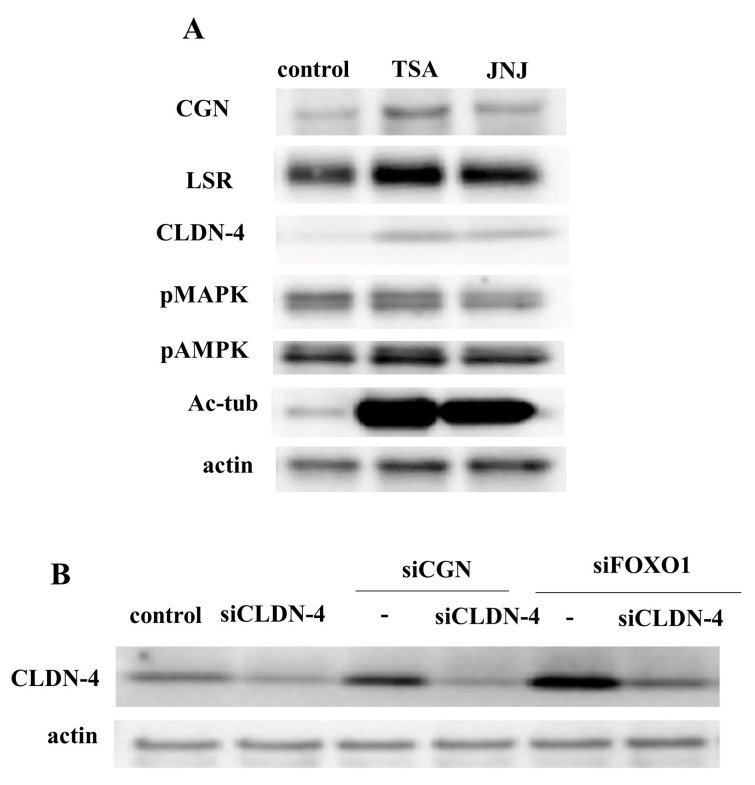
(**A**) Western blotting for angulin-1/LSR, CGN, CLDN-4, pMAPK, pAMPK, Ac-tubulin, and actin in normal human lung epithelial cells (HLE cells) treated with the HDAC inhibitors TSA and Quisinostat at 10 μM. (**B**) Western blotting for CLDN-4 and actin in HLE cells with the knockdown of CLDN-4 and CGN or FOXO1.

**Figure 7 ijms-25-01411-f007:**
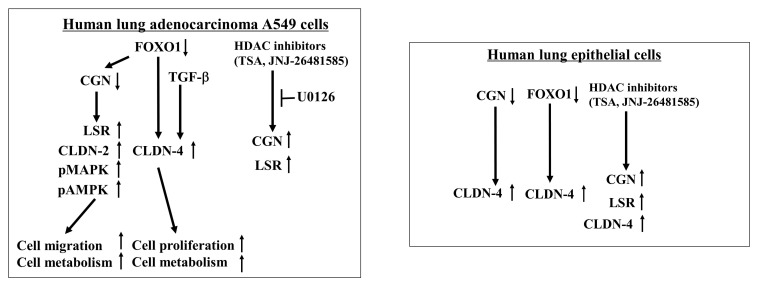
Overview of interactions and roles of CGN, FOXO1, CLDN-2, and CLDN-4 in human adenocarcinoma A549 cells and normal lung epithelial cells. ↑: upregulation. ↓: downregulation.

## Data Availability

Data are contained within the article and Appendix A.

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
