# Peer review of "The Roles and Regulatory Mechanisms of Tight Junction Protein Cingulin and Transcription Factor Forkhead Box Protein O1 in Human Lung Adenocarcinoma A549 Cells and Normal Lung Epithelial Cells"

_ijms, 2024, doi:10.3390/ijms25031411_

Round 1
Reviewer 1 Report
Comments and Suggestions for Authors
In this manuscript entitled “The roles and Regulatory Mechanisms of Tight Junction Protein Cingulin and Transcription Factor Forkhead Box Protein O1 in Human Lung Adenocarcinoma A549 Cells and Normal Lung Epithelial Cells”, the authors investigated the Histone deacetylase (HDAC) inhibitors affect expression of both CGN and FOXO1 to regulating the development of lung cancer. The following are some issues that should be addressed by the authors.
(1)This paper explores the expression and distribution of CGN in lung adenocarcinoma samples, and should be added investigate changes in the distribution and expression of FOXO1 during carcinogenesis of human lung adenocarcinoma.
(2)I noticed some significant issues that could affect the credibility and relevance of your study. First of all, the article lacks data of clinical samples. This may lead to insufficient data and make it difficult to draw clear conclusions. In lung cancer research, a larger sample size is typically required to ensure the reliability and representativeness of the data. I suggest increasing validation at the clinical level,to enhance the statistical power of your study.
(3)The functional experiments in this paper mainly focus on the effects of CGN and FOXO1 on lung cancer cells, but there are many related studies on these two genes, and a group of experiments on the effects of Histone Deacetylase (HDAC) inhibitors on cell function should be added at the same time.
(4)The manuscript contains several spelling and grammar errors. I kindly request a thorough review and careful editing of the main text.
(5)The quality of all figures presented in the manuscript appears suboptimal. For example, the expression of reference protein in Figure 2A and B;Figure 4 A. is quite different, so it is impossible to prove your conclusion as an accurate reference.
Comments on the Quality of English Language
The manuscript contains several spelling and grammar errors.
Author Response
(1)This paper explores the expression and distribution of CGN in lung adenocarcinoma samples, and should be added investigate changes in the distribution and expression of FOXO1 during carcinogenesis of human lung adenocarcinoma.
We agreed with your suggestions. However, Since FOXO1 was not detected in normal lung tissues and lung adenocarcinoma tissues by the antibodies which we used in the present study, we did not indicate the changes in the distribution and expression of FOXO1 during carcinogenesis of human lung adenocarcinoma.
It is known that FOXO1 is downregulated in human NSCLC tissues. We added the information in Introduction.
(2)I noticed some significant issues that could affect the credibility and relevance of your study. First of all, the article lacks data of clinical samples. This may lead to insufficient data and make it difficult to draw clear conclusions. In lung cancer research, a larger sample size is typically required to ensure the reliability and representativeness of the data. I suggest increasing validation at the clinical level,to enhance the statistical power of your study.
We agreed with your suggestions. We first in the world indicated the changes in expression and localization of CGN during carcinogenesis in lung adenocarcinoma by using clinical samples. In the present study, we mainly indicated how CGN and FOXO1 contribute to the malignancy of lung adenocarcinoma by using lung adenocarcinoma cell line and normal lung cells.
(3)The functional experiments in this paper mainly focus on the effects of CGN and FOXO1 on lung cancer cells, but there are many related studies on these two genes, and a group of experiments on the effects of Histone Deacetylase (HDAC) inhibitors on cell function should be added at the same time.
We know that there are some related studies on FOXO1 in lung cancer and HDAC inhibitors and FOXO1 is involved in human lung carcinogenesis. However, the studies related on CGN are rare in lung adenocarcinoma and normal lung cells. Furthermore, although it is known that HDAC inhibitors induced some tight junction proteins, the effects of HDAC inhibitors against CGN are unknown in lung adenocarcinoma and normal lung cells.
(4)The manuscript contains several spelling and grammar errors. I kindly request a thorough review and careful editing of the main text.
We checked them.
(5)The quality of all figures presented in the manuscript appears suboptimal. For example, the expression of reference protein in Figure 2A and B;Figure 4 A. is quite different, so it is impossible to prove your conclusion as an accurate reference.
We performed all experiments at several times. We indicated the best images.
Comments on the Quality of English Language
The manuscript contains several spelling and grammar errors.
We checked them.

Reviewer 2 Report
Comments and Suggestions for Authors
The manuscript "The roles and Regulatory Mechanisms of Tight Junction Pro-2 tein Cingulin and Transcription Factor Forkhead Box Protein 3 O1 in Human Lung Adenocarcinoma A549 Cells and Normal 4 Lung Epithelial Cells" is a well written and informative article highlighting the activity of HDAC inhibitors and the role of CNG and FOXP1. Here is my comment on the manuscript:
1. The Figure 1 has IHC results represented. All the images if quantified would give a better understanding of the localization and expression of the Cingulin.
2. Another set of results to justify the activity of FOXO1 in human lung adenocarcinoma similar to how CNG is established would give a connection as to why both the factors are studied here in the same manuscript. Maybe some IHC would add enough to the manuscript. Also I think the result section needs to be modified that way where a connection to why FOXO1 is studied can be justified.
Author Response
- The Figure 1 has IHC results represented. All the images if quantified would give a better understanding of the localization and expression of the Cingulin (CGN).
We added the quantification in the localization and expression of CGN in Figure 1E.
- Another set of results to justify the activity of FOXO1 in human lung adenocarcinoma similar to how CGN is established would give a connection as to why both the factors are studied here in the same manuscript. Maybe some IHC would add enough to the manuscript. Also I think the result section needs to be modified that way where a connection to why FOXO1 is studied can be justified.
It is known that loss of FOXO1 results in altered organization of tight junction protein occludin in intestine (Chen et al., 2021).
(Chen Z, Luo J, Li J, Kim G, Chen ES, Xiao S, Snapper SB, Bao B, An D, Blumberg RS, Lin CH, Wang S, Zhong J, Liu K, Li Q, Wu C, Kuchroo VK. Foxo1 controls gut homeostasis and commensalism by regulating mucus secretion. J Exp Med. 2021, 218(9), e20210324.)
However, the relationship between FOXO1 and CGN is unknown in lung.
Forkhead box protein (FOXO1) is downregulated in human NSCLC tissues and silencing of FOXO1 promotes proliferation, migration, and invasion of NSCLC cells in vitro, whereas overexpression of FOXO1 inhibits the migration and invasion.
(Gao, Z.; Liu, R.; Ye, N.; Liu, C.; Li, X.; Guo, X,; Zhang, Z.; Li, X.; Yao, Y.; Jiang, X. FOXO1 inhibits tumor cell migration via regulating cell surface morphology in non-small cell lung cancer cells. Cell Physiol Biochem. 2018, 48(1), 138-148.)
Expression of some tight junction proteins contributes to EMT.
Accordingly, we investigated the relationship between CGN and FOXO1.

Reviewer 3 Report
Comments and Suggestions for Authors
In their study titled “The Roles and Regulatory Mechanisms of Tight Junction Protein Cingulin and Transcription Factor Forkhead Box Protein O1 in Human Lung Adenocarcinoma A549 Cells and Normal Lung Epithelial Cells,” Daichi Ishii and colleagues explore the impact of CGN and FOXO1 on the progression of non-small cell lung cancer (NSCLC). They discover that reducing CGN levels through FOXO1 increases NSCLC malignancy. The research also highlights the potential of histone acetylase inhibitors, specifically TSA and quisinostat, in treating lung adenocarcinoma by altering CGN and FOXO1 expression. The topic of this manuscript is interesting; however, the presentation of the rough data and overall data quality impacts the manuscript's credibility and the persuasiveness of the findings. I have several comments and concerns to address.
1.Ensure that all abbreviations are defined at their first use. For example, "CLDN-2", "CLDN-4", "MAPK/AMPK pathways", and "angulin-1/LSR" should be explained if they haven't been defined earlier in the text.
2.It would be beneficial if the authors could provide more detailed information regarding the Ethics Statement, such as the specific ethics approval number. In this study, the authors investigate the expression and distribution of the target protein in both cancerous and normal lung tissues. Regarding the 'normal tissues', do they represent the tissue adjacent to the cancerous areas? It should be claimed clearly.
3.In Figure 1, it would be beneficial if the authors could magnify certain areas to better illustrate the expression and distribution of the target protein between the normal lung tissue and non-small cell lung cancer tissue.
4.In Figure 2C, could the image be rotated from vertical to horizontal orientation to improve the overall appearance of the panel?
5.It appears that actin, used as an internal control in the Western blot assay, exhibits unstable expression and may not be suitable for this role.
6.Several font sizes in the figures are too small to read. Please standardize the font size and type across all figures for better readability.
7.In Figure 2B, it would be beneficial for clarity and accuracy to add error bars to the statistical histogram, representing the variability in protein expression. Additionally, could the authors clarify how the levels of target proteins were quantified from the Western blot bands, and how many replicates were performed in these experiments?
8.In the Migration Assay section, it's crucial to understand whether gene knockdown or inhibitor treatment impacts cell growth and proliferation. How do the authors mitigate the influence of cell growth and proliferation on the results of the Migration Assay?
9.In Figure 5, please include the statistical differences between the control and experimental groups. Additionally, it would be helpful to know the number of repeats performed in this assay. Also, the error bars on the statistical histograms appear to be quite large, suggesting high variability.
10.Why did the authors choose to focus exclusively on A549 cells without validating their findings in other cell lines simultaneously?
Comments on the Quality of English LanguagePlease check the manuscript carefully
Author Response
- Ensure that all abbreviations are defined at their first use. For example, "CLDN-2", "CLDN-4", "MAPK/AMPK pathways", and "angulin-1/LSR" should be explained if they haven't been defined earlier in the text.
We added the abbreviations at their first use.
- It would be beneficial if the authors could provide more detailed information regarding the Ethics Statement, such as the specific ethics approval number. In this study, the authors investigate the expression and distribution of the target protein in both cancerous and normal lung tissues. Regarding the 'normal tissues', do they represent the tissue adjacent to the cancerous areas? It should be claimed clearly.
We used the tissues around adenocarcinoma tissues as normal lung tissues and added the sentence.
- In Figure 1, it would be beneficial if the authors could magnify certain areas to better illustrate the expression and distribution of the target protein between the normal lung tissue and non-small cell lung cancer tissue.
We changed Figure 1E (delated title) and added the quantification.
- In Figure 2C, could the image be rotated from vertical to horizontal orientation to improve the overall appearance of the panel?
We changed the figure 2C.
- It appears that actin, used as an internal control in the Western blot assay, exhibits unstable expression and may not be suitable for this role.
We used actin bands as an internal control in the Western blot assay. As in Figure 2B left, the actin bands were not good, we changed to the actin bands of one more experiment.
- Several font sizes in the figures are too small to read. Please standardize the font size and type across all figures for better readability.
We changed font sizes and type in the almost figures.
- In Figure 2B, it would be beneficial for clarity and accuracy to add error bars to the statistical histogram, representing the variability in protein expression. Additionally, could the authors clarify how the levels of target proteins were quantified from the Western blot bands, and how many replicates were performed in these experiments?
We agree with the referee’s suggestions. We performed two times in Figure 2B and indicated the average in the graphs.
- In the Migration Assay section, it's crucial to understand whether gene knockdown or inhibitor treatment impacts cell growth and proliferation. How do the authors mitigate the influence of cell growth and proliferation on the results of the Migration Assay?
We rewrote the detailed method.
At 48 h after the transfection with siRNAs, a wound was created in the cell layer using a plastic pipette tip (P100), and then at 8 h after wound healing, the length of the wound was subsequently measured using a microscope imaging system (Olympus, Tokyo, Japan). Since we measured at 8 h after wound healing, the cell growth and proliferation did not affect cell migration assay
- In Figure 5, please include the statistical differences between the control and experimental groups. Additionally, it would be helpful to know the number of repeats performed in this assay. Also, the error bars on the statistical histograms appear to be quite large, suggesting high variability.
We rewrote the sentences of results in Figure 5.
Knockdown of CGN increased cell metabolism, evident from changes in baseline oxygen consumption rates (OCR), maximal OCR, spare respiratory capacity (SRC), and ATP production, while knockdown of FOXO1 increased baseline OCR (Figure 5A, 5B, 5C, 5D).
- Why did the authors choose to focus exclusively on A549 cells without validating their findings in other cell lines simultaneously?
A549 cells are a useful model as lung adenocarcinoma and many studies are reported. We previously reported the roles of another tight junction proteins angulin-1/LSR and claudin-2 in human lung adenocarcinoma by using A549 cells.
Comments on the Quality of English Language
Please check the manuscript carefully
We checked them.

Reviewer 4 Report
Comments and Suggestions for Authors
The authors did a thorough investigation on the regulation of the histone deacetylase (HDAC) inhibitors on cingulin (CGN) and transcription factor forkhead box protein O1 (FOXO1) levels in non-small cell lung cancer (NSCLC) cell line A549 and primary normal lung cells. Overall, the manuscript did a comprehensive investigation regarding the roles of CGN and FOXO1 in NSCLC cell survival and activity and will attract wide interest in this field. I suggest minor revisions. Here are some comments to the authors.
1. I would suggest to add a graphical illustration at the beginning of the manuscript to describe the cell pathway/mechanism of the CGN and FOXO1 (something like Figure 7).
2. Missing scale bar in Figure 1C and 1D.
3. Please include statistical analysis for Figures 2A and 2B.
4. In section 2.6, it’s hard to make the conclusion as the error bars in the results are big to represent differences upon treatment.
5. Is there any cell survival data or corresponding in vivo study done by the authors or in the literature? It would be great to include more application directions in the discussion section.
Author Response
- I would suggest to add a graphical illustration at the beginning of the manuscript to describe the cell pathway/mechanism of the CGN and FOXO1 (something like Figure 7).
We added a graphical illustration.
- Missing scale bar in Figure 1C and 1D.
We added scale bars in Figure 1C and 1D.
- Please include statistical analysis for Figures 2A and 2B.
We performed statistical analysis and added it.
- In section 2.6, it’s hard to make the conclusion as the error bars in the results are big to represent differences upon treatment.
We rewrote the results indicated Figure 5C and 5D.
- Is there any cell survival data or corresponding in vivo study done by the authors or in the literature? It would be great to include more application directions in the discussion section.
We added The Cancer Genome Atlas (TCGA) data as supplemental data in the discussion.

Round 2
Reviewer 1 Report
Comments and Suggestions for Authors
The manuscript can be accepted for publication.
Author Response
The manuscript can be accepted for publication.
Thank you for your efforts.

Reviewer 3 Report
Comments and Suggestions for Authors
Please label the changes in the manuscript, as some comments appear to be unaddressed despite the author's assertion in the rebuttal letter that they have been resolved.
Comments on the Quality of English LanguageMinor editing of English language required
Author Response
Please label the changes in the manuscript, as some comments appear to be unaddressed despite the author's assertion in the rebuttal letter that they have been resolved.
We are sorry for the rebuttal letter. We agree with your comments and suggestions and labeled the changes in the manuscript.
Comments on the Quality of English Language
Minor editing of English language required.
We checked it by the native speaker.

Round 3
Reviewer 3 Report
Comments and Suggestions for Authors
Thank you for the authors' response. The revised manuscript has shown some improvement, but there are still several concerns and comments that need to be addressed before considering submission.
1. The Western blotting bands of actin have not been replaced with better quality ones. Most of the actin bands are inconsistent across different experimental groups, despite the protein being used as an internal control.
2. The figures need modifications to enhance their professional appearance and eliminate any amateurish aspects. This should include adjustments to the layout, aspect ratio, font size and type, along with the labels for the horizontal and vertical coordinates.
Comments on the Quality of English LanguageMinor editing of English language required
Author Response
- The Western blotting bands of actin have not been replaced with better quality ones. Most of the actin bands are inconsistent across different experimental groups, despite the protein being used as an internal control.
We miswrote the comments to the referee. The new image of actin bands in Figure 2B left is detected by using same sample.
- The figures need modifications to enhance their professional appearance and eliminate any amateurish aspects. This should include adjustments to the layout, aspect ratio, font size and type, along with the labels for the horizontal and vertical coordinates.
We are sorry for our amateurish aspects and indicated the best layout as good as possible.
Comments on the Quality of English Language
Minor editing of English language required.
We checked it by the native speaker.

Round 4
Reviewer 3 Report
Comments and Suggestions for Authors
Thank you for the authors' prompt response. I am pleased to note that all of my comments and concerns have been addressed satisfactorily.